# Research on Small Acceptance Domain Text Detection Algorithm Based on Attention Mechanism and Hybrid Feature Pyramid

**Mingzhu Liu** *⊕, **Ben Li** ⊕ and **Wei Zhang**

The Higher Educational Key Laboratory for Measuring & Control Technology and Instrumentation of Heilongjiang Province, Harbin University of Science and Technology, Harbin 150080, China
* Correspondence: lmz@hrbust.edu.cn

**Abstract:** In the traditional text detection process, the text area of the small receptive field in the video image is easily ignored, the features that can be extracted are few, and the calculation is large. These problems are not conducive to the recognition of text information. In this paper, a lightweight network structure on the basis of the EAST algorithm, the Convolution Block Attention Module (CBAM), is proposed. It is suitable for the spatial and channel hybrid attention module of text feature extraction of the natural scene video images. The improved structure proposed in this paper can obtain deep network features of text and reduce the computation of text feature extraction. Additionally, a hybrid feature pyramid + BLSTM network is designed to improve the attention to the small acceptance domain text regions and the text sequence features of the region. The test results on the ICDAR2015 demonstrate that the improved construction can effectively boost the attention of small acceptance domain text regions and improve the sequence feature detection accuracy of small acceptance domain of long text regions without significantly increasing computation. At the same time, the proposed network constructions are superior to the traditional EAST algorithm and other improved algorithms in accuracy rate P, recall rate R, and F-value.

**Keywords:** lightweight network structure; hybrid feature pyramid; BLSTM network; acceptance domain; attention mechanism

## 1. Introduction

In the last several years, the evolution of cultural exchanges and network technology has become faster and faster, and considerable video resources, such as movies, documentaries, and various shows, have sprung up on the Internet and have begun to affect people's lives. Due to cultural differences and personal understanding differences, there are still many text messages that the audience has difficulty noticing and understanding, such as street signs, trademarks, and commodities in videos, even though there are subtitles in these videos. If this information can be extracted, recognized, and translated, it will be helpful for the audience to understand the video information. Therefore, text detection in natural scenes is of great research significance.

Directly applying target detection technology to natural scene text detection is the most direct way in early natural scene text detection. Among them, the representative ways are Faster-RCNN [1] and SSD [2]. However, Faster-RCNN and SSD still have low efficiency for long text and slanted text detection due to the complexity of natural scene text detection, such as different types, unfixed position and arrangement direction, and the diversity of size and shape [3]. Thus, researchers consider improving the construction of the original target detection algorithm, making it more suitable for natural scene text detection.

Methods of traditional text detection and some text detection methods based on deep learning mostly use multi-stage constructions. Each level needs to be adjusted and optimized during training. This inevitably affects the final text location effect and

is very time-consuming. To solve the above problems, Kuangshi Technology Company proposed an Efficient and Accurate Scene Text Detector algorithm (EAST) in 2017 [4]. The EAST algorithm comes up with an end-to-end text detection method, which can directly forecast the text line region and eliminate multiple stages in the middle, such as candidate region aggregation, text segmentation, and post-processing. The construction of the EAST model is simple. Since it adopts FCN full convolution neural network construction and non-maximum suppression (NMS) merging process, the accuracy of text detection on ICDAR2015 can reach 80.57% [5]. Furthermore, the EAST algorithm is no longer limited to text size and text proportion and can detect any size and scale of text because it abandons the idea of anchor in Faster-RCNN and directly forecasts text lines [6]. Additionally, the EAST algorithm uses locality-aware NMS to filter the generated geometry, contributing to reducing the probability of text boxes being deleted falsely [7]. The EAST algorithm can forecast two different forms of the text box, namely slanted text and irregular text, which cannot be detected by CTPN and other algorithms [8]. However, the EAST algorithm does not pay enough attention to the small acceptance domain text area and does not deal with the small acceptance domain feature image, which will increase the difficulty of follow-up processing [9]. To solve the problem of insufficient attention to the small acceptance domain, based on the EAST algorithm, combined with the characteristics of text in natural scenes, this paper improves the feature extraction network, optimizes the feature pyramid construction, and completes the text detection.

## 2. Preliminaries

The EAST algorithm is divided into two stages: full convolution network (FCN)for forecasting text boxes and local perception non-maximum suppression (LNMS). The network construction of the FCN (see Figure A1 in Appendix A), where $f_i$ ($i$ is 1, 2, 3, 4), represents the output feature map of different size feature pyramids, and $h_i$ denotes the output feature map of the fusion of different acceptance domains. The network construction can be decomposed into three parts: Feature extractor, Feature merge, and Output [10]. In the first stage, the image is inputted into the FCN network construction, then the single-channel pixel-level text fractional feature map and multi-channel geometric feature map are generated. Generally, the text area is marked by two geometric shapes: a rotating box and a horizontal box [11]. The EAST algorithm also sets the corresponding loss function for each geometric shape and puts a threshold into each prediction region, and the geometry whose score exceeds the threshold is considered effective. Afterward, the result processed by LNMS is taken as the final detection result of the output [12].

The original EAST algorithm uses VGG16 as the feature extraction network [13]. Although VGG16 is quite mature, it has only 16 tiers, otherwise, a mass of parameters is needed for the calculation during the training. Therefore, Google launched the latest lightweight network MobileNetV3 in May 2019 [14]. The new version of Mobile Net adopts more new features, such as the introduction of $5 \times 5$ depth convolution into MobileNetV3 to replace the original $3 \times 3$ depth convolution, and the introduction of a squeeze-and-excitation (SE) [15] module and h-swish (HS) [16] activation function to improve the model accuracy. As revealed from the comparison of VGG16 construction between MobileNetV3 and EAST original algorithm, the parameter of VGG16 is 138 million, while the parameter of MobileNetV3 is only 0.054 million, which accounts for only 4% of the VGG16 network parameter. The computation of VGG16 is 15.3 billion, while that of MobileNetV3 is only 219 million, which accounts for only 1.4% of the VGG16 network. Simultaneously, the accuracy of MobileNetV3 is 75.2%, which is higher than that of VGG16 (71.5%). Mobile Net network has great merits in lightweight neural networks, such as being smaller in size, a smaller computation, and higher accuracy. In this paper, based on VGG, deep separable convolution is used to take the place of the ordinary convolution of the traditional VGG network, and the attention mechanism is integrated into the VGG network to obtain a more efficient feature extraction effect.

In MobileNetV3 construction, deep Separable Convolution (Depth-wise Convolution) is used [17]. It is constituted of a tier of Depth-wise convolution and a tier of Point-wise convolution. Each tier of convolution is followed by batch normalization and ReLU activation function. The difference between deeply separable convolution and standard convolution is that the parameters and the computation are significantly reduced when the accuracy remains unchanged. For the contrast between depth-separable convolution and traditional convolution, see Figure A2 in Appendix A.

## 3. Methods

### 3.1. Lightweight Network Structure with a Mixed Attention Mechanism

SE module is introduced in MobileNetV3, and it is a kind of attention mechanism [18]. SE block is a sub-construction that can be embedded in other classification or detection models rather than a complete network construction.

#### 3.1.1. Channel Domain Attention Mechanism

This attention mechanism is divided into three sections: Squeeze, Excitation, and Attention [19]. SE module construction see Figure A3 in Appendix A.

The squeeze function in the squeeze construction as in Equation (1) [19]:

$$z_C = F_{sq}(u_c) = \frac{1}{H \times W} \sum_{i=1}^{H} \sum_{j=1}^{W} u_c(i,j) \tag{1}$$

where $u_c(i,j)$ represents the feature value of the $i$-th row and $j$-th column of the $c$-th channel in the feature map $u$; $H$ and $W$ denotes the length and width of the feature map severally; $z_C$ indicates the output of the $c$-th channel. The squeeze function shown in Equation (1) completes a global average operation, namely global average pooling.

The excitation function in the excitation construction as in Equation (2) [19]:

$$s_c = F_{ex}(z_c, W) = \sigma(g(z_c, W)) = \sigma(W_2 \frac{1}{1 + e^{-W_1 z_c}}) \tag{2}$$

where $zc$ is the squeezing function shown in Equation (1). $\sigma$ is the ReLU function; the dimensions of $W_1$ and $W_2$ are $C/r \cdot C$ and $C \cdot C/r$, where $r$ is a scaling parameter. The purpose of introducing parameter $r$ is to decrease the number of accesses, thus decreasing the total computation; $r$ is set to 16 in this paper. By training and learning the weight values of $W_1$ and $W_2$, a one-dimensional weight $sc$ is acquired to activate each tier channels which is shown in Equation (2). The final output $\widetilde{X}_C$ of the block is obtained by the scale function as in Equation (3) [19]:

$$X_c = F_{scale}(u_c, s_c) = s_c \cdot u_c \tag{3}$$

where $s_c$ is the output of the $c$-th feature map of the excitation function, and $u_c$ is the output feature value of the $c$-th channel of the feature map. As shown in Equation (3), by multiplying the eigenvalues of disparate channels with disparate weights, the key access weight is large and the non-key channel weight is small, which can enhance the attention to the key channel domain.

#### 3.1.2. Spatial Domain Attention Mechanism

Another kind of attention mechanism is the attention mechanism of the space domain, and its construction (see Figure A4 in Appendix A). According to the feature graph obtained from the former processing, the maximum value is globally pooled on the channel, and the average value is globally pooled on the dimension. Then, the feature information of different locations in the feature graph is compared and extracted, and the weight of each location is acquired. The output $O(i,j)$ of the aftermath of the maximum global pooling as in Equation (4) [18]:

$$O(i,j) = \max(I_m(i,j)), m \in (1, C) \tag{4}$$

where $I_m(i,j)$ indicates the feature value of the feature map at the $(i,j)$ position on the $m$-th channel, $C$ is the total channel number of the input feature map.

The average global pooling handle as in Equation (5) [18]:

$$P(i,j) = \frac{1}{d}\sum_{n=1}^{d} T_n(i,j) \tag{5}$$

where $T_n(i,j)$ represents the feature value at the location $(i,j)$ of the $n$-th access, and $d$ is the number of the amount of accesses in the input feature map. The two feature maps are linked in the access dimension utilizing the access link, then a $1 \times 1$ convolution kernel is used for convolution operation. Finally, a sigmoid function is used to process an output. This attention mechanism can effectively draw the important message of each location of the feature map and the main features by assigning weights and suppressing the background information.

### 3.1.3. Hybrid Attention Mechanism Module

In the two attention mechanisms, the attention in the spatial domain neglects the message in the access domain, and the image features in each access are treated equally. This will make the spatial domain operation limited to the original image feature extraction stage, resulting in the application of the neural network in each tier of the interpretability to be not strong. On the other hand, the attention of the channel domain is to pool the overall situation, which means to obtain information directly and to ignore the local news in every access. Therefore, a lightweight hybrid-domain attention mechanism model is constructed by combining the characteristics of the two attention mechanisms and can perform attention in access and spatial dimensions [20]. The construction of the Convolution Block Attention Module (CBAM) is illustrated in Figure 1.

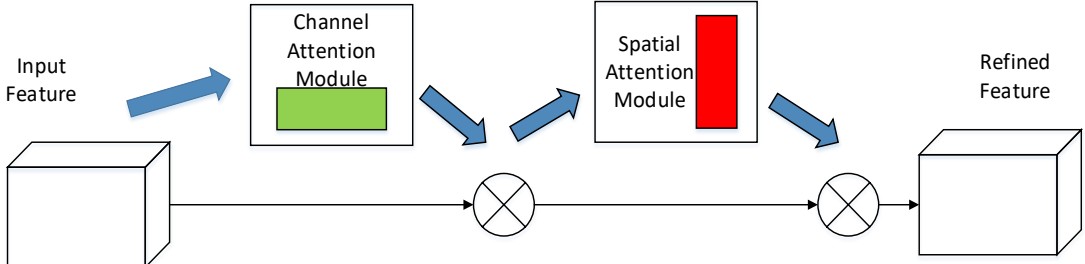

**Figure 1.** CBAM construction diagram with the hybrid attention mechanism module.

As mentioned above, CBAM is made up of two sub-modules; one is the Channel Attention Module (CAM), and the other is the Spatial Attention Module (SAM). CBAM is to multiply the input feature map with the CAM to get a new feature map, then multiplies the new feature map with SAM to get the resulting output. Thus, the channel domain and the spatial domain are combined. Because SE and CBAM belong to the access domain attention module and hybrid domain attention module, respectively, CBAM has a better attention concentration effect than the SE module. Therefore, this paper uses the attention mechanism module CBAM to replace the SE module in the lightweight network MobileNetV3. At the same time, combined with ResNet-50, this paper deepens the number of network tiers to 50 to achieve the purpose of enriching high-level semantic information [21].

### 3.2. Optimization Way of Small Acceptance Domain Based on a Hybrid Feature Pyramid

In the FCN network construction, the EAST algorithm adopts a four-tier feature pyramid construction and merges in the feature fusion part. Through deconvolution operation, the small-scale feature map is expanded to the same size as the large-scale feature map, and then the access dimensions are connected by adding in the channel domain. From the FCN construction (see Figure A1 in Appendix A), it can be seen that in the feature merging process, the small acceptance domain feature map at the bottom of the feature extraction network is merged first. This construction will lead to the small acceptance domain feature map only [22] through a little convolution calculation, and when the number of accesses is small, the proportion of fused information is less [22]. It will lead to the problem that the EAST algorithm does not pay enough attention to the small acceptance domain text, and the small acceptance domain text feature map is less concerned and operated, which will not be conducive to its feature extraction. For the sake of solving the problem of the small acceptance domain, a hybrid feature pyramid construction is applied to optimize feature collection and feature merging [23].

As shown in Figure 2, the hybrid feature pyramid before optimization contains top-down and bottom-up paths, and then the two paths are combined. A path with the same direction as the original EAST algorithm and a path with the opposite direction are fused as the final output. In this method, the latter small acceptance domain feature map is located at the bottom of the feature merging. Thus, the problem of insufficient attention to the small acceptance domain existing in the feature pyramid construction of the original EAST algorithm is solved through the two-path merging. On the other hand, in the original feature pyramid construction, too much attention is paid to the large acceptance domain, which leads to too much feature extraction of the large receptive feature map. Through the test, it is found that when the two-way path is used to merge the large acceptance domain regions, the detection accuracy is not significantly improved, but it will bring the burden of calculation. Therefore, the hybrid feature pyramid construction is redesigned in this paper. The new pyramid construction proposed in this paper only merges the first two tiers with a smaller acceptance domain of the original feature pyramid, then merges with the path in the other direction. The specific optimization construction is shown in Figure 3. The dotted box in Figure 3 shows the construction of the acceptance domain optimization part in the improved feature pyramid. The feature map of the new feature pyramid construction still comes from the output of the feature extraction network. The two feature extraction paths of the new construction and the original construction are almost the same, as shown in Figure 2, except that the path direction of the feature merging part is different, so it not only solves the problem that the EAST algorithm does not pay enough attention to the text in the acceptance domain but also increases the computation that can be ignored.

However, in the EAST algorithm, enough attention has been paid to the large acceptance domain so that the features of the large receptive feature graph are rich enough. Therefore, the feature detection accuracy of the large receptive feature graph is not significantly improved by the bi-directional path combination but will bring computational burden. Therefore, the mixed-feature pyramid construction is redesigned in this paper. Only the first two tiers of the feature pyramid are merged (the two tiers with smaller acceptance domains) on the merge path in the opposite direction, and then the fusion is carried out with the path in the other direction. The concrete construction is shown in Figure 3 (the dotted box is the optimized part). Compared with the original construction, the improved construction retains the characteristics of the original construction, but the improved construction is more concise and lightweight. Due to the features of the merger figure coming from the output of the network of feature extraction, two feature extraction paths are the same (according to the Figure 2, feature extraction is conducted only once), only in part are features combined in a different direction. Therefore, the improved construction will only bring a slight increase in the computation during the feature combination of the first two tiers. Taking the 50-tier feature extraction network as an example, at least 50 convolution operations are required during the extraction and 5 times during the feature combination.

Moreover, the number of channels in the feature map of the small acceptance domain are less, and the computation during the combination is lower than that of the large acceptance domain combination. Therefore, the calculation amount of the final parameter number is less than 1/50 of the original algorithm, while the original mixed feature pyramid will bring about 1/10 of the computation. So, the optimized mixing characteristics of the pyramid sense retains the original construction with an upgrade to small wild features. It solves the sense that there is too little wild text, and there are less acceptance domain characteristics graph processing in feature extraction because the characteristics of the pyramid before two tiers merge will not bring a burden for the network of the computation.

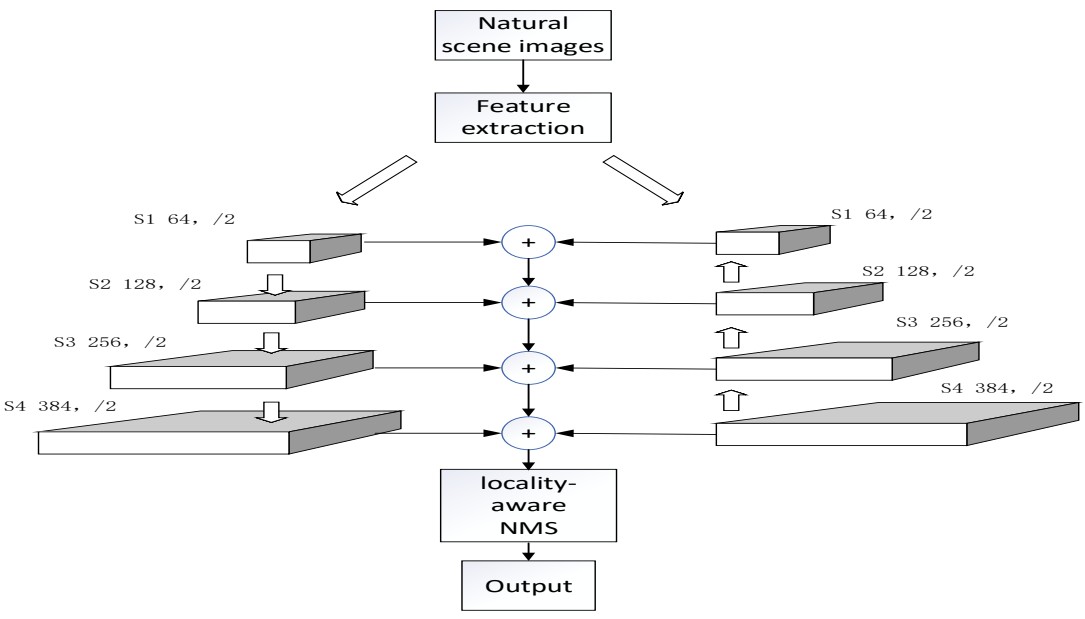

**Figure 2.** Schematic diagram of the original hybrid feature pyramid construction.

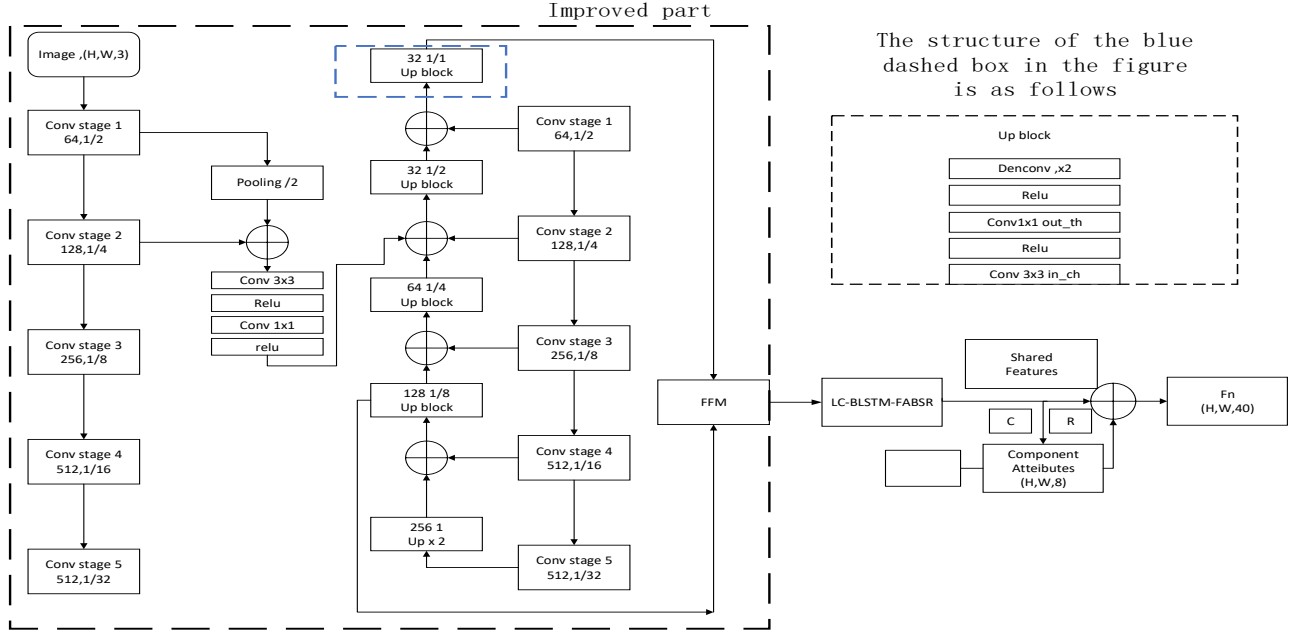

**Figure 3.** Construction block diagram of the improved algorithm.

At present, to solve the problem of the unbalanced distribution of targets, most methods adopt balanced sampling and hard negative mining. Although this will improve the

network performance, this technology inevitably incorporates a stage and more parameters to adjust the pipeline. This is contrary to the design of the EAST algorithm. To simplify the training process, equilibrium cross-entropy (the same as solving class imbalance, $\beta$ is the number of counterexamples/total number of samples) is used in this paper, and the formula as in Equation (6):

$$L_{\text{s}} = balanced - xent(\hat{Y}, Y^*) = -\beta Y^* \log \hat{Y} - (1 - \beta)(1 - Y^*) \log(1 - \hat{Y}) \tag{6}$$

$\hat{Y}$ is the prediction of the fractional graph and $Y^*$ is the labeled value. $\beta$ is the balance factor between positive and negative samples as in Equation (7):

$$\beta = 1 - \frac{\sum y^* \in Y^* * y^*}{|Y^*|} \tag{7}$$

$L_g$ geometry loss is divided into two parts, one part is $IoU$ loss, and one part is rotation angle loss as in Equations (8) and (9):

$$L_{AABB} = -InIoU(\hat{R} \cap R^*) = -\ln \frac{\hat{R} \cap R^*}{\hat{R} \cup R^*} \tag{8}$$

$$L_\theta(\hat{\theta}, \theta^*) = 1 - \cos(\hat{\theta} - \theta^*) \tag{9}$$

Among them, $\theta$ is the prediction of the rotation Angle, and $\overset{*}{\theta}$ Indicates the annotation value. Finally, the overall geometric loss is the sum of the $AABB$ loss and the angular loss, the formula is as in Equation (10):

$$L_g = L_{AABB} + \lambda_\theta L_\theta \tag{10}$$

Feature Extraction of Text Sequence Based on BLSTM Network

Attention mechanism includes not only spatial–domain and channel–domain attention mechanisms mentioned above, but also the time–domain attention mechanism. This attention mechanism focuses on a special feature: a sequence feature usually called context information, which is the unique feature information of the text. Because the traditional CNN network construction cannot effectively integrate the sequence features, this paper adds the BLSTM network to the conventional feature extraction CNN algorithm [24]. The BLSTM network construction is shown in Figure 4a, where $x_n$, $y_n$ and $s_n$ represent the input sequence, output sequence, and sequence characteristics at different times, respectively. The construction of memory cell A in BLSTM is shown in Figure 4b, which is similar to the LSTM network. It is a kind of special construction with three gates: the forgetting gate, the input gate, and the output gate. With these three gates, LSTM can more effectively determine which information is forgotten or retained. LSTM has the ability of forwarding propagation and it can easily extract and output long-distance text features, so the LSTM network can effectively solve the learning of long sequence information. The BLSTM network construction is a combination of two LSTM network constructions in different directions. Context information can be integrated well by combining the two different propagation directions. The improved CTPN text detection algorithm proposed in this paper adopts the improved CNN-BLSTM network construction and combines the characteristics of a text sequence, which makes the detection results more robust to the long-term text. However, due to the existence of feature pyramid construction in the EAST algorithm, the BLSTM network cannot be directly added to the feature extraction CNN network construction. Therefore, this paper researches the specific way of adding a BLSTM network to feature fusion and output tier. It can improve the precision of the network and also obtain preferable test results. The comparison effect of text detection with or without BLSTM network construction is shown in Figure 5.

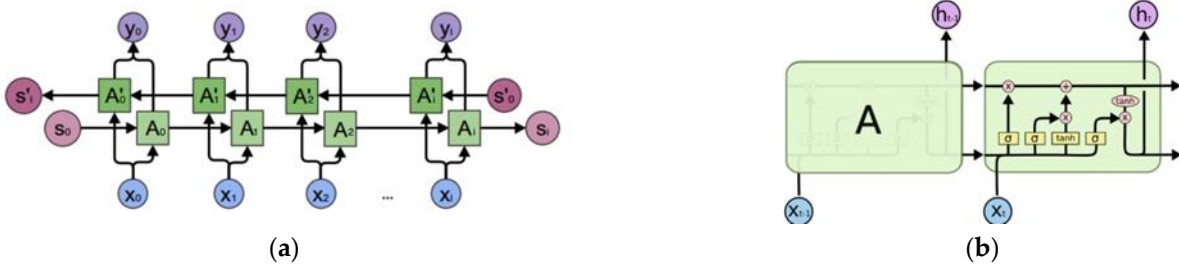

(**a**)                                (**b**)

**Figure 4.** BLSTM network construction. (**a**) Schematic diagram of BLSTM network construction. (**b**) Construction diagram of Memory cell A in the BLSTM.

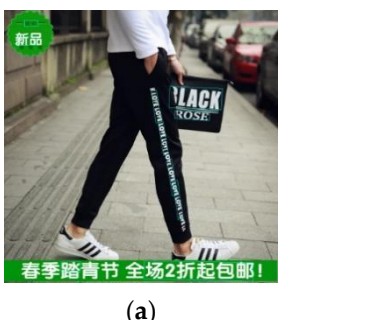    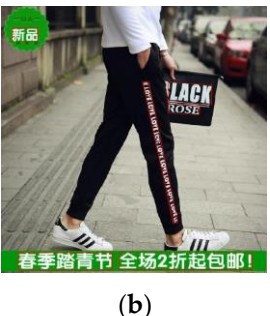

(**a**)                                (**b**)

**Figure 5.** Comparison of text detection effect with or without adding BLSTM network. (**a**) Test result without BLSTM network; (**b**) Test result with BLSTM network.

The results shown in Figure 5 show that for the traditional algorithm without adding BLSTM network construction, due to the lack of sequence features, the algorithm cannot detect whether the text belongs to the same sentence. Therefore, when detecting long text, the long text will be detected into multiple short texts by mistake, and some texts will not even be detected, as shown in Figure 5a. After the addition of the BLSTM network, long texts are detected well owing to the integration of sequence features, thus contributing to the improvement of the detection accuracy, as shown in Figure 5b.

## 4. Experimental Results

### 4.1. Test Description

For the sake of testing the effectiveness of the new construction of a hybrid attention mechanism, hybrid feature pyramid, and BLSTM network for text detection in natural scenes, this paper not only tests but also analyses the hybrid feature pyramid performance and ablation performance of the new algorithm, as well as the text detection performance of the algorithm combined with a hybrid attention mechanism.

The framework used in the test is TensorFlow1.12 in the pre-training model. The test uses the improved lightweight network MobileNet-50 and uses the ADAM optimizer to drill the network from end to end. For the sake of expediting the learning, $512 \times 512$ frame rate images are evenly sampled from the images. After rotation and translation, the initial batch size is set to start training from 16. The learning rate of the Adam optimizer starts from $10^{-3}$, and decays to one-tenth of the original after every 1000 training, until the end of training when it reaches the optimal level.

Among many data sets, the ICDAR2015 data set which includes one thousand training images and five hundred test images is especially suitable for text training and detection in natural scenes. In this paper, the ICDAR2015 data set is selected as the training set and the test set, and the test results will be compared. For the sake of verifying the effectiveness of the new construction of BLSTM and CBAM, the ablation experiment was carried out on the ICDAR2015 data set, and the precision rate P, the recall rate R, and F-value were used as assessment indexes.

### 4.2. Comprehensive Comparison of Several Natural Scene Text Detection Algorithms

The proposed algorithm is compared with other algorithms such as Seglink, TextSnke, and RRPN on the ICDAR2015 dataset. Meanwhile, this paper selects the video frame image with natural scene text in the video stream to test the algorithm and the test results are shown in Figures 6–8. In these figures, it can be observed that the result of the algorithm based on the optimization of the hybrid feature pyramid has a better performance than that of the traditional EAST arithmetic, because the former can attract more attention to the small acceptance domain so it has a better detection effect on the text area of the small acceptance domain. However, whether it is slanted text, slanted perspective text, long sequences text, or complex background text, the improved algorithm in this paper can achieve a better text detection effect than the other two algorithms. Especially for small acceptance domain text, the text detection effect of the proposed algorithm is particularly outstanding.

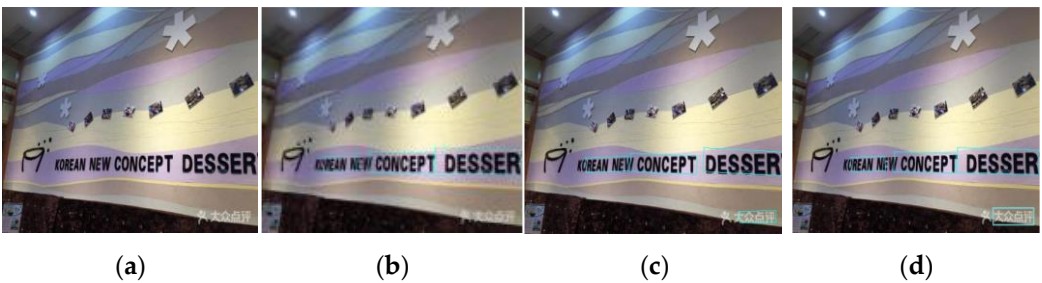

| **(a)** | **(b)** | **(c)** | **(d)** |

**Figure 6.** Comparison of the detection effect of slanted text. (**a**) Slanted original text; (**b**) EAST algorithm; (**c**) Hybrid feature pyramid algorithm; (**d**) The algorithm of this paper.

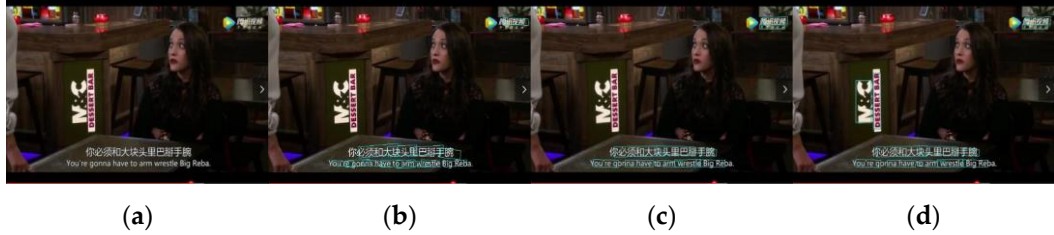

| **(a)** | **(b)** | **(c)** | **(d)** |

**Figure 7.** Comparison of the detection effect of long sequences text. (**a**) Original long text; (**b**) EAST algorithm; (**c**) Hybrid feature pyramid algorithm; (**d**) The algorithm of this paper.

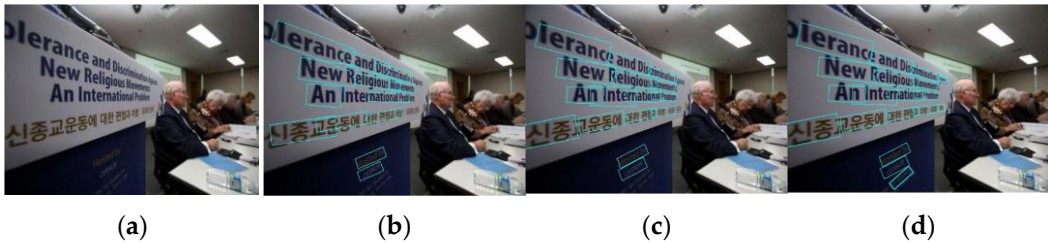

| **(a)** | **(b)** | **(c)** | **(d)** |

**Figure 8.** Comparison of the detection effect on complex natural background text. (**a**) Original complex background text; (**b**) EAST algorithm; (**c**) Hybrid feature pyramid algorithm; (**d**) The algorithm of this paper.

Table 1 lists the accuracy P, the recall rate R, and the F-value performance of the CTPN algorithm, the Seglink algorithm, several improved East algorithms proposed in other literature, and the improved algorithm proposed in this paper in text detection. The results show that the F-value of the proposed algorithm can reach 83%, and the recall rate R can reach 86%, which is higher than other text detection algorithms such as Seglink, TextSnke, and RRPN. The accuracy rate P of the improved algorithm proposed in this paper reaches 80%, which is higher than most text detection algorithms, but slightly lower than TextSnke and RRPN algorithms. As shown in Table 1, the improved algorithm proposed in this text

has obvious advantages in text detection. Compared with the original EAST algorithm, the recall rate R and the accuracy rate P of the improved construction are improved by 8% and 6%, respectively, and the F-value is improved by 7%. Therefore, the proposed optimization construction has obvious performance advantages compared with other improved algorithms and constructions.

**Table 1.** Comparison results of R, P, and F-value evaluation indexes of several algorithms.

| Algorithm Name | R (%) | P (%) | F (%) |
|---|---|---|---|
| EAST + VGG16 | 72.8 | 80.5 | 75.4 |
| Seglink [25] | 76.8 | 80.5 | 75 |
| CTPN + VGG16 [26] | 51.6 | 74.2 | 60.9 |
| EAST + resnet50 + BLSTM [27] | 78.07 | 85.10 | 81.64 |
| EAST + PVANET2 × MS [28] | 77.23 | 84.64 | 80.77 |
| TextSnke [29] | 83.20 | 73.90 | 78.72 |
| RRPN [30] | 86.00 | 70.00 | 77.00 |
| The algorithm in this paper | 80.02 | 86.04 | 82.93 |

## 5. Conclusions

An improved EAST algorithm for text detection is proposed in this paper. The complications of text localization and inaccurate detection and high calculation load in the traditional network construction can be handled by adopting a hybrid attention mechanism method and introducing a lightweight feature extraction neural network construction. The method in this paper can reduce the computation, deepen the number of tiers of the network, and improve the accuracy of the network. Meanwhile, the improved hybrid attention mechanism is combined with the hybrid pyramid model to solve the problem of deficiency attention to small acceptance domain text and insufficient feature extraction in the EAST algorithm. Furthermore, the BLSTM network construction is also added after the feature fusion to effectively integrate the sequence features of the text region and make the text detection more robust. The test results demonstrate that the improved EAST algorithm presented in this paper has better performance than other existing ways of detecting text regions in natural scenes and complex scenes, and significantly reduces the number of parameters and calculations.

Therefore, the improved algorithm proposed in this paper is indeed a good algorithm. However, this study has three limitations. First, in the process of testing the algorithm, this study has not found a better training method for the improved algorithm, and the training method adopted in this paper is still based on the traditional network model, although this training method can also have a certain effect on the accuracy of text recognition under complex backgrounds. However, if we can find a training model that is more compatible with the lightweight network, it will help to further improve the recognition accuracy and improve the problems of insufficient attention to the text in small receptive fields and insufficient feature extraction. Second, through a large number of experimental tests, it is concluded that the EAST algorithm proposed in this paper, compared with other algorithms in terms of recognition accuracy, improved and can reduce the amount of calculation. However, the improved algorithm computational complexity of the theory of quantitative analysis model is constructed. In the future, we will further build the calculation theory of quantitative analysis model and verify its effectiveness. Finally, in the experimental test of this paper, a special data set for text detection and recognition (ICDAR2015) is used. Although this data set is widely applicable to the training and detection of conventional text, it also has a shortage of a single background scene. Therefore, because of the research field of complex background text recognition in this paper, it is necessary to collect and sort out more complex scene text data sets, to provide more reliable and richer model training resources for text detection and recognition in complex scenes.

**Author Contributions:** Conceptualization, M.L.; methodology, M.L. and B.L.; software, B.L.; validation, B.L. and W.Z.; formal analysis, M.L.; investigation, M.L. and B.L.; resources, M.L. and B.L.; data curation, B.L.; writing—original draft preparation, M.L. and B.L.; writing-review and editing, M.L. and B.L.; visualization, B.L. and W.Z.; supervision, M.L. All authors have read and agreed to the published version of the manuscript.

**Funding:** This research received no external funding.

**Data Availability Statement:** The data presented in this study are available on request from the corresponding author.

**Conflicts of Interest:** The authors declare no conflict of interest.

## Abbreviations

| Short Name | Full Name |
| --- | --- |
| CBAM | Convolution Block Attention Module |
| EAST | Efficient and Accurate Scene Text |
| NMS | non-maximum suppression |
| FCN | full convolution network |
| LNMS | local perception non-maximum suppression |
| SE | squeeze-and-excitation |
| HS | h-swish |
| CAM | Channel Attention Module |
| SAM | Spatial Attention Module |
| BLSTM | bidirectional long short-term memory |
| ICDAR | International Conference on Document Analysis and Recognition |
| Faster-RCNN | Faster-Region-Convolutional Neural Network |
| CNN | Convolutional Neural Network |
| SSD | Single Shot MultiBoxDetector |
| CTPN | Connectionist Text Proposal Network |
| VGG | Visual Geometry Group |
| ReLU | Rectified Linear Unit |
| ResNet | Deep residual network |
| IoU | Intersection over Union |
| LSTM | Long Short Term Memory |
| ADAM | Adaptive Moment |
| RRPN | Rotation Region Proposal Network |

**Appendix A**

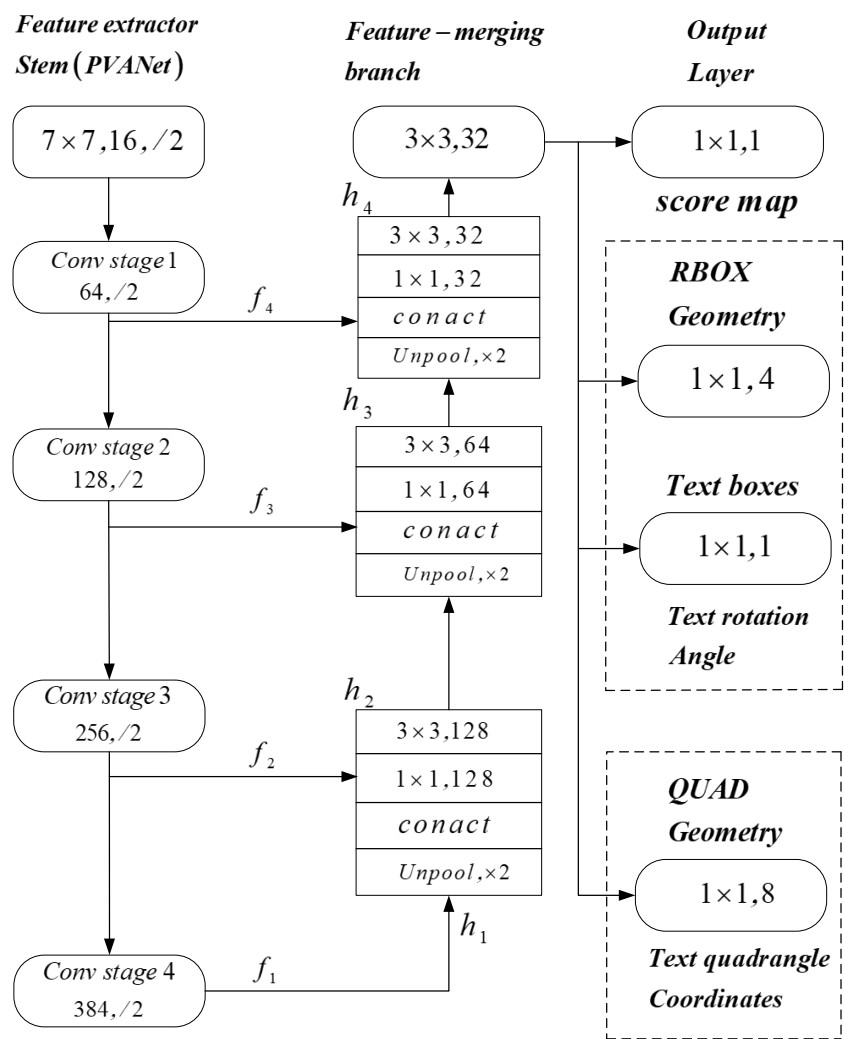

**Figure A1.** School diagram of FCN network construction.

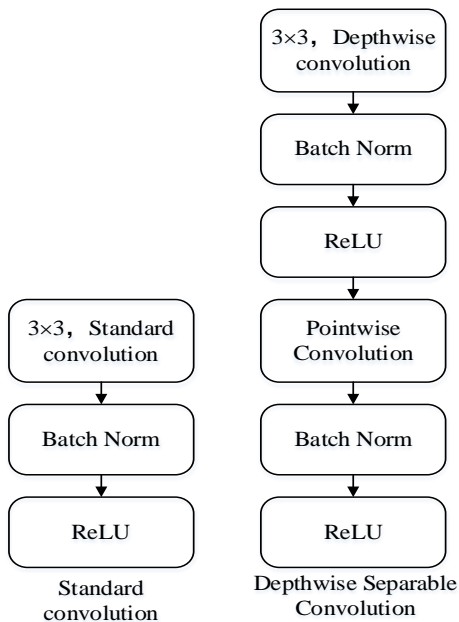

**Figure A2.** Standard separable convolution and Depth wise separable convolution.

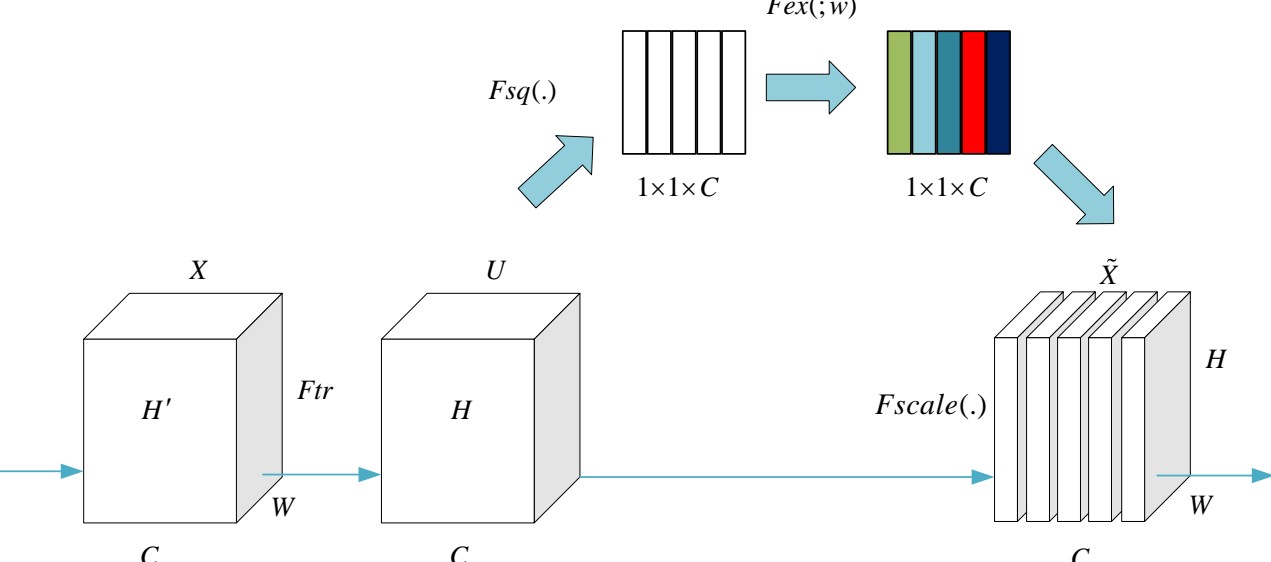

**Figure A3.** Schematic diagram of SE module construction.

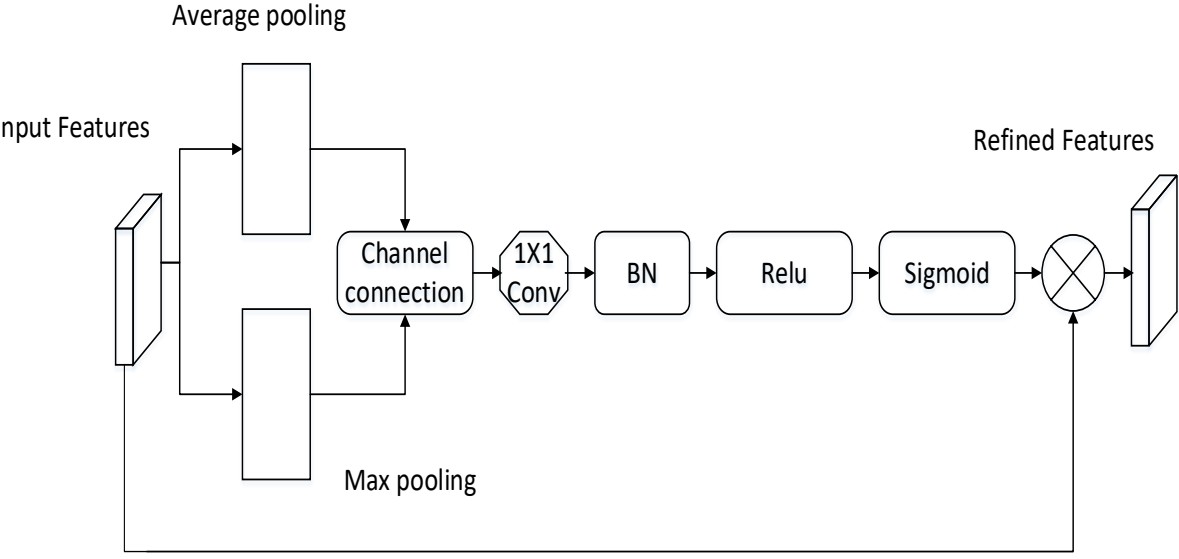

**Figure A4.** Schematic diagram of spatial attention mechanism.

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
