# Peer review of "Research on Small Acceptance Domain Text Detection Algorithm Based on Attention Mechanism and Hybrid Feature Pyramid"

_electronics, doi:10.3390/electronics11213559_

Round 1
Reviewer 1 Report
Dear Authors,
Please accept sincere congratulations for your paper!
Still, you'll find some suggestions in the original message to the editorial team (bellow):
<<
Honored Editorial Board,
As the title, abstract, content and conclusions suggest, this paper (id electronics-1971629) proposes a light-10 weight network structure on the base of the EAST algorithm, which is Convolution Block Attention 11 Module (CBAM). The authors claim that the proposed network constructions are superior to other algorithms in terms of accuracy.
After reading the entire paper, I think I found that there are still some issues waiting to be dealt with.
I will start with the format ones and then I will continue with those related to the paper’s content and substance if applicable.
I mention below the following:
-
English language and style issues - Grammarly (https://app.grammarly.com) on default settings (American English, Set Goals: Audience=Knowledgeable, Formality=Neutral, Domain=General) detected only for the text block resulting from the concatenation of Title+Abstract+Keywords+Conclusion:
(a) 9 correctness issues / critical alerts (change prepositions, definite articles, correct choice of words, agreement mistakes, etc.)
and
(b) 17 more advanced ones, namely: Passive voice misuse (6), Word choice (4), Unclear sentences (3), Punctuation in compound/complex sentences (2), Intricate text (1), and more (1).
The resulting Grammarly's overall score was 76 (fair but still not good) out of 100 (max) for this four-component sample above.
Still, since none of the authors appears to be a native English speaker, I suggest a total revision of the English language and style for the entire article using Grammarly or another specialized tool;
-
The paper should precisely follow the specific structure of the journal, namely:
Author Information, Abstract, Keywords, Introduction, Materials & Methods, Results, Discussion, Conclusions, etc., as clearly indicated at: https://www.mdpi.com/journal/electronics/instructions; -
The links from the cited references (e.g., [1] [2] ..[20] ) in the main text to the References section are missing;
-
All references to all equations or formulas must be explicitly and precisely formulated in the main text (e.g. “see eq. N”);
-
Figures 8-12 clearly suffer in terms of resolution when compared to others (minimum 1000 pixels width/height, or a resolution of 300 dpi or higher according to the Journal’s instructions at the link above). The authors should at least scale them up a little bit and check the image compression options when exporting to .pdf;
-
Some figures are vertically distorted (e.g., Figure 3);
-
The authors must avoid ending some sections/subsections with figures or other components (e.g. Figure 2 just before section 3); The authors are also required to check for similar issues in the whole manuscript;
-
Some subsections are isolated from their titles (each on different pages - e.g., 3.2, line 140). The authors are also required to check for similar issues in the whole manuscript;
-
Overall, there are so many figures (12) included in the main content. A special section (Appendix - at the end of the manuscript) must be included and those figures considered by the authors not essential for understanding the main ideas must be moved there and properly renumbered (e.g., see Figure A1 in Appendix A);
-
The plural “conclusions” is needed because there is more than a single conclusion in the corresponding paragraph;
-
More, the authors should provide a list of abbreviations (e.g., BLSTM, CNN, LSTM, etc.) at the end of the manuscript;
-
When introducing formulas / equations, the authors are required to also include some references to papers in indexed journals mentioning such/similar constructs;
-
The authors are required to include more explanations and precise details about the common view of the accuracy values (>=70% and <80%-fair models; >=80% and <90%-good models; >=90%-very good/excellent models) and its application here. They should provide more references to scientific papers where this topic is considered and the accuracy intervals are precisely defined;
-
The authors must understand that replicability in science (https://doi.org/10.1007/s10516-021-09610-2 ) is not a fad but a necessity. Therefore, they should insert a Data Availability Statement at the end of the manuscript and include there all precise links to all data providers’ datasets;
-
In the same spirit of replicability support ( https://doi.org/10.1038/nature.2016.20504 ), if some custom algorithms have been used (e.g., Figure 7), the authors must precisely identify them in their own GitHub repository. If not existing, the authors must create one for the entire project corresponding to this manuscript;
-
Moreover, the authors must precisely specify details about the hardware and software tools they used (Data and Method section);
-
The final list of just 20 references, suggests that the Literature Review/Related Works part needs more development. Actually, I think that much more contributions in journal papers must be cited in this research both in the first part of the paper and especially in the section dedicated to the interpretation of the results;
-
I also think that a section regarding the limitations of the methodology used is completely missing from this manuscript.
Thank you for the opportunity to read and check this contribution!
>>
I wish you all the best!
Reviewer 2 Report
the contents of paper are well defined. the authors need to check the formatting of paper properly.
The figures are not properly aligned.
The author must also incorporate the time complexity of system and its implications regarding real-time implementation.
Round 2
Reviewer 1 Report
Dear Authors,
You performed some improvements to the manuscript.
I think your paper is now closer to the state of being published.
I wish you all the best!